# MRNG: Accessing Cosmic Radiation as an Entropy Source for a Non-Deterministic Random Number Generator

**DOI:** 10.3390/e25060854

**Published:** 2023-05-26

**Authors:** Stefan Kutschera, Wolfgang Slany, Patrick Ratschiller, Sarina Gursch, Håvard Dagenborg

**Affiliations:** 1Institute of Software Technology, Graz University of Technology, 8010 Graz, Austria; 2Department of Computer Science, UiT the Arctic University of Norway, 9037 Tromsø, Norway

**Keywords:** privacy-enhancing technologies, entropy source, wireless and mobile security and privacy, computer engineering, cryptography, cyber security, information security, privacy, security, software engineering

## Abstract

Privacy and security require not only strong algorithms but also reliable and readily available sources of randomness. To tackle this problem, one of the causes of single-event upsets is the utilization of a non-deterministic entropy source, specifically ultra-high energy cosmic rays. An adapted prototype based on existing muon detection technology was used as the methodology during the experiment and tested for its statistical strength. Our results show that the random bit sequence extracted from the detections successfully passed established randomness tests. The detections correspond to cosmic rays recorded using a common smartphone during our experiment. Despite the limited sample, our work provides valuable insights into the use of ultra-high energy cosmic rays as an entropy source.

## 1. Introduction

Randomness is fundamental for many security primitives, tools, and services. For instance, smartphone users might require a facility to transmit, receive, or transport sensitive and confidential information, necessitating robust security measures for data protection. Furthermore, secure communication requires endpoints to pick random encryption keys and use random nonces to fight replay attacks. Databases that process sensitive personal data can better protect the privacy of individuals by adding random noise to aggregated query responses [1]. Although a strong notion of differential privacy [2] can be achieved in such systems using only a weak source of randomness [3,4], Garfinkel and Leclerc [5] argue for the need of a strong, secure, reliable, and abundant source of entropy, such as one that can be created using quantum mechanics Brask et al. [6]. *Pseudorandom Number Generators* (PRNGs) are widely accessible; however, obtaining random numbers that rely on true randomness, as for instance, the *Quantis* random number generator [7], is significantly more challenging.

Building upon our previous work [8,9], this paper describes a novel method for using *Ultra High Energy Cosmic Rays* (UHECR) captured by an image sensor in common smartphones as a cheap, robust, and readily available source of true randomness. We propose an algorithm that can extract randomness out of UHECR and cosmic radiation on a smartphone to be embedded it into an *Random Number Generator* (RNG). Although such an RNG is slow, it provides a strong and cost-effective randomness that is suitable for any system in need of locally generated true random numbers. To the best of our knowledge, this is the first work that successfully accesses cosmic radiation as an entropy source on a common smartphone.

We implemented a prototype to show how a random bit generator can be constructed in practice using cosmic radiation as an entropy source and to further evaluate which methods are suitable for extracting randomness of sufficient strength to pass standardized tests. Our prototype builds on the existing *Cosmic Ray Extremely Distributed Observatory* (CREDO) Android application developed by Homola et al. [10], which we have extended with our proposed algorithm and various backend communication optimizations. Our adapted application then collects images of UHECR and muons. On each interference between UHECR and the blackened *Complementary Metal Oxide Semiconductor* (CMOS)/*Charge Coupled Device* (CCD) image sensor, a resulting shift in the pixel color triggers the application to capture the detection.

In summary, our work makes the following contributions:We show that random numbers can be extracted from a yet unused entropy source for randomness, namely, UHECR.We describe four methods that, in combination, extract randomness out of UHECR, as discussed in Section 4.2.2.Our proposed *Muon Random Number Generator* (MRNG) prototype (named in honor of the research on muons collected among UHECR) does not need any external services or devices in order to extract randomness from UHECR. Our MRNG prototype works on any Android smartphone with an API level of 14 or higher, including Android 4.0, which was released in 2011. This is stated in more detail in Section 4.2.Most importantly, we have proved that our extracted random sequence, out of the UHECR entropy source, is truly random when tested against NIST SP.800-22 statistical test suite. This is stated within Section 6.Furthermore, we may have accidentally discovered a new splash-like representation of (presumably) UHECR, as discussed and shown in Section 7.1.

## 2. Threat Model

A smartphone user may want to access or communicate sensitive, confidential information and data through an application. Methods such as diplomatic luggage codes, secret keywords sent in distress, and encryption may be feasible for a broader spectrum of individuals. In order to encrypt information and data, respectively, RNGs are often used. The encryption is in supposed danger when an attacker can reproduce the secret seed (PRNGs) or predict the random sequence by limiting the entropy through design flaws or eavesdropping on the RNG.

## 3. Background

During the Middle Ages, randomness out of dice and astragali were attributed to supernatural forces and were used to predict the future [11,12]. In his famous quote about quantum mechanics, Einstein states that *”god does not play dice“*. Whereas Poincare [13] differentiates between random phenomena, where calculus provides insight into how a phenomenon works, and random phenomena, where the laws behind them are not yet fully discovered [8,9].

Batanero et al. [11] have a similar viewpoint to Poincare [13], as the authors argue that randomness exists in two groups, namely formal and informal. In their work, informal is used in order to describe currently unexplained topics, and formal describes the lack of predictability or patterns. The lack of predictability or patterns can also be viewed in terms of the Kolmogorov complexity [14]. Some examples of informal randomness are, for instance, *Buffon’s Needle*, *Buffon’s Noodle*, or the *Monte Carlo* simulation, because at first glance, it seems to be a random yet magical approximation of the mathematical constant π, but it can be precisely explained mathematically [9,15,16,17,18].

### 3.1. Random Number Generators

The term deterministic is used when the output for each input into a system always stays the same. In other words, a system is surjective and can be called deterministic when each value in the ’input’ set is always paired with the same value of the ’output’ set. Such RNGs are known as PRNGs or deterministic-*Digital Random Number Generators* (DRNGs). In contrast, Barker and Kelsey [19] stated that a non-deterministic RNG has access to a full entropy source. The full entropy source can produce an ideal random sequence, and each produced bit is unpredictable and unbiased. Therefore, the values are also independent of the other bits in the sequence. An ideal random sequence of *n* bits contains *n* bits of entropy [19]. Such RNGs are known as *True Random Number Generators* (TRNGs) or non-deterministic-RNGs. In summary, a system can be called non-deterministic when the output of each input is always different, and it can be called deterministic when the same input always results in the same output.

The DRNG is a hardware-based RNG that utilizes thermal noise as a non-deterministic entropy source [20]. This entropy source, as described by Garfinkel and Leclerc [5], outputs random numbers as bits at 3 GHz.

There are three categories of *Quantum Random Number Generators* (QRNGs), namely a trusted device, self-testing, and semi-self-testing, as analyzed by Ma et al. [21]. The quantum physical mechanisms are exploited in order to generate random bits at a usually high rate. The methods of using quantum physics, without going into detail, include the detection of the state of a qubit, where the polarization of the photon plays an important role, or the measurement of the spatial respectively temporal mode of a photon. Another approach by Sanguinetti et al. [22] is to resolve the photon number distribution of a light-emitting diode with a smartphone, as cited in Ma et al. [21].

In addition to these definitions above, it can be said that the cascade construction is a typical architecture of RNGs Mechalas [20], Bellare et al. [23].

### 3.2. Relevant Existing Random Number Generators

In this section, we discuss the most relevant RNGs. The majority of them apply a modulo operation on the entropy source to generate random bits.

#### 3.2.1. RNG Proposed by Park

Park et al. [24] built a TRNG that extracts randomness from the dark noise of the CMOS image sensor. The data are post-processed using techniques such as applying a Hankel matrix and a universal hashing function. The TRNG can generate a random sequence with up to 2.4 Mb/s; it was successfully tested using NIST SP.800-22 statistical test suite and evaluated for entropy with NIST SP.800-90B on at least $10^{6}$ samples [24,25,26].

#### 3.2.2. RNG Proposed by Zhang

Zhang et al. [27] proposed a TRNG based on placing a human finger on a smartphone camera. The authors argue that using the flashlight to illuminate the finger causes unpredictable patterns. The resulting data stream was then processed with the (mod2) operation. The authors claimed to have successfully extracted entropy-enhanced true random bits. However, they also noted that the Bayer mosaic pattern and the associated post-processing demosaicing process could negatively impact randomness [27]. The authors did not provide any bit rate.

#### 3.2.3. RNG Proposed by Leschiutta

Leschiutta [28] developed an RNG known as ’BlueRand’ with the aim of sending encrypted messages within arbitrary messaging services [28,29]. The BlueRand scheme involves creating a one-time pad by applying a bitwise shift to the least significant bit of the blue color channels of two independent images; this is done through a (mod2) operation. The RNGs performance was analyzed using ENT and RaBiGeTe [28,29], but no bit rate was provided.

#### 3.2.4. RNG Proposed by Chen

The RNG involves extracting coordinates from video input and setting a threshold for future values at a specific coordinate [30]. The RNG then applies a bitwise operation on the base RGB color channels, correlating them with the sampled audio. The authors reported that the RNG passed all tests in the NIST SP.800-22 statistical test suite [30]. The authors did not provide any bit rate.

#### 3.2.5. RNG Proposed by Krhovják

Krhovják et al. [31] proposed an RNG based on the average entropy within test objects, using Nokia N73, E-Ten X500, and E-Ten M700 as test subjects. The authors described four methods: processing raw values only, using the least significant color bit, combining colors with XOR, and employing flip-flop bit extraction. The authors argued that using (mod2) on each pixel for bit extraction was the most robust method. The RNG reached a bit rate of at least 13.85 KB/s. However, the output of this method did not pass all tests in the NIST SP.800-22 statistical test suite [31].

#### 3.2.6. RNG Proposed by Reezwana

Reezwana et al. [32] implemented and tested a QRNG on a nano-satellite in low Earth orbit, called SpooQy1 [33]. They tested their RNG with excellent results within a laboratory setting. The executed experiment in the low Earth orbit consisted of 66 trials, where each pass gained 256 bits of random data. A sequence of 16,896 bits is too short for it to be tested with the DieHarder test suite. Nonetheless, the authors managed to test for the Borel normality criterion [34], where a lack of patterns was indicated, thus suggesting a random sequence.

### 3.3. Random Number Test Suites

An essential part involves testing the randomness and unpredictability of the acquired data. We chose the NIST SP.800-22 statistical test suite due to its ability to handle smaller sample sequences within the test execution setup. The NIST SP.800-22 statistical test suite consists of 15 tests. However, most tests within the NIST SP.800-22 statistical test suite require large sequences to perform the tests [35]. Although, the NIST SP.800-22 statistical test suite is subject to critics [36] and may change soon [37], it is still the current state-of-the-art test suite in security and cryptography, respectively.

The full NIST SP.800-22 statistical test suite only applies to large sequences n≥1 000 000 bits. Nonetheless, each test is documented in detail and states a minimum input for *n*. Tests with a recommendation of n≥ 100 or n≥ 128 are listed below.


(NIST Test 2.1) The Frequency (Monobit) Test;
n≥ 100
(NIST Test 2.2) Frequency Test within a Block;
n≥ 100
(NIST Test 2.3) The Runs Test;
n≥ 100
(NIST Test 2.4) Tests for the Longest-Run-of-Ones in a Block;
n≥ 128
(NIST Test 2.12) The Approximate Entropy Test; 10.0 cm
n≥ 100
(NIST Test 2.13) The Cumulative Sums (Cusum) Test.
n≥ 100

Therefore, some research focuses on evaluating and testing the randomness and unpredictability of short sequences. Sulak et al. [38] propose an alternative approach to determining randomness for 512-bit length short sequences. Future work and publications building on the findings of Sulak et al. [38] propose further tests.

Furthermore, Reezwana et al. [32], Martínez et al. [39] suggest the Borel normality criterion by Calude [34] to indicate a lack of patterns within a tested sequence. The Borel normality criterion allows for an asymptomatic test on even short sequences with respect to the correlating Borel normality criterion level, allowing for the possibility of indicating the lack of patterns within the tested sequence and implying a suggestion for a random sequence.

The most recent results of Li and Lin [40] propose a first interval check algorithm for short binary sequences. Furthermore, statistical values are calculated and compared with the NIST SP.800-22 statistical test suite. Future work on the test for short sequences needs to be adapted.

It can be summarized that it is best to test smaller sample sizes with the NIST SP.800-22 statistical test suite and correctly adapt the parameters to fit the needs [40].

### 3.4. Cosmic Radiation

Primary cosmic radiation hits the Earth’s upper atmosphere continuously. By interfering with the Earth’s atmosphere, the primary cosmic radiation particles, such as protons and alpha particles, decay into photons, leptons, electrons, positrons, pions, and muons. The outcome of the decay process is known as secondary radiation. Muons travel with a velocity of 29.8 cm ns^−1^, which is 99.4 % of the speed of light (0.994·c); they have a mean lifetime of 2.197 μs and a half-life of 1.523 μs [41,42]. It can be assumed that there is one muon hit within an area of 1 cm${}^{2}$ every minute. The detection rate for muons doubles with every 1500 m increase in altitude. Muons are interesting for several reasons. They not only prove Einstein’s special relativity theories, namely time dilation and length contraction [43,44], but can also interact with semiconductor material, despite having weak interference properties [45,46,47].

Radiation emitted by the sun, or solar radiation, may also contain particles that interfere with the semiconductor material built into electronic devices. The sun has a natural 11-year solar cycle, in which the peaks in the solar radiation flux can be predicted with relative accuracy. Nonetheless, unpredictable coronal mass ejections or geomagnetic storms, respectively, still occur. Examples of extreme events are the Carrington Event 1859, August 1972, and October 1989. This is also shown with the CREME96-model [48].

#### 3.4.1. Single Event Effect

Any interference of a single incident particle that can be measured is called the *Single Event Effect* (SEE). The SEE can be further divided into destructive and non-destructive categories. *Single Event Latch-up* (SEL) and *Single Event Burnout* (SEB) are destructive events, whereas *Single Event Transient* (SET) and *Single Event Upset* (SEU) are non-destructive [49,50,51]. However, their causes may not always be related to cosmic radiation, as shown by Brazil [52].

##### SEU Incidents

A thoroughly analyzed incident, which was most likely caused by the SEU, left 110 of 303 persons on an airplane injured, 12 of them with serious injuries, on Qantas flight 72 from Singapore to Australia. The report by the Australian Transport Safety Bureau [53], with reference number AO-2008-070, states that a UHECR may have triggered a SEE by interfering with one of the integrated circuits within the CPU module. Whereas other incidents, such as the addition of 4096 votes to a candidate in a 2003 election in Belgium, as discussed in the article by Bhuva [54], and the recorded ’glitch’ during a speedrun of the video game Super Mario 64, as discussed by Brazil [52], have been reported.

The effect on SEUs within large-scale *Field Programmable Gate Array* (FPGAs) systems was discussed by [55]; they found that an SEU can be expected every 3.75 h within a 100,000 FPGAs system. Such errors most likely caused the acceleration problem within Toyota vehicles [56,57].

In addition to the direct impact of SEU on individuals, 152 parity bit errors were recorded by the supercomputer *Cray-1* during the first 6 months after installment [50] in 1976. Due to the parity bit errors, the *Error Correcting Code* (ECC) mechanism was implemented. Almost 3 decades later in 2002, a new supercomputer named *Q* faced an unpredicted amount of errors. Engineers, already aware of SEUs, discovered that the *Static Random Access Memory* (SRAM) of *Q* was only covert by the parity error check but not ECC [50]. Already aware of SEUs, it was discovered that the SRAM of *Q* only had a parity error check but no ECC [50].

#### 3.4.2. Smartphone-Based Cosmic Ray Detectors

As cited by Homola et al. [10], there are many detection methods for cosmic radiation, such as scintillator-based particle detectors, air fluorescence detectors, or water-based Cherenkov detectors; however, in this research, we are particularly interested in CMOS/CCD image sensors. These can be found frequently in modern-day smartphones. When charged particles or radiation, such as gamma or X-rays, travel through a smartphone’s image sensor, they have a similar effect on the image sensor as light does. By covering the camera lens and, therefore, forcing a near-black image, it is possible to make passing particles visible during their interference with the CMOS/CCD image sensor. Such representations of particles on interference can be seen in the first figure in Section 5.

The CREDO collaboration [10], among other projects, such as ‘*Cosmic RAYs Found In Smartphones* (CRAYFIS)’ [58] or *Distributed Electronic Cosmic-ray Observatory* (DECO) [59], have implemented applications of many different kinds, as described by Whiteson et al. [58], Vandenbroucke et al. [59], Bibrzycki et al. [60], Niedźwiecki et al. [61] and the Cosmic Ray app [62]. Furthermore, it was possible to migrate the application to laptops and computers of various operating systems with the CREDO web page implementation. Currently, global research on *Cosmic Ray Ensembles* (CRE) has led to the development of projects focused on creating smartphone-based cosmic ray detectors; for instance, *DECO* [59], *CRAYFIS* [58], *CREDO* [10], and the *Cosmic Ray app* [62].

According to Bar et al. [63], there are four different types, namely spots, tracks, worms, and artifacts; the latter are unwanted data often caused by misuse or cheating attempts. The remaining three types, spots, tracks, and worms, are being researched within the CREDO project [10,60,63]. The definition of different types of radiation and their presence on an image has also been discussed by [59], the creators behind the *DECO* project.

The primary focus of the research conducted by Homola et al. [10] and Whiteson et al. [58], respectively, is not only to distinguish between the different types of radiation, but also to research *Primary Cosmic Ray* (PCR) arrays, the cause of PCR, and *Extensive Air Showers* (EAS) caused by PCR. Interestingly, EASs are the causes of secondary decay processes in the generation of muons [64].

## 4. Experiment

### 4.1. Experiment Setup

For the implementation of the MRNG prototype [65] the CREDO Android application [66] was adopted. The MRNG prototype was installed on 8 devices used for the test. The devices were laid out on a table attached to a permanent power supply within a building. Since muons are reluctant to interfere with physical matter, this was an appropriate location. All devices had their camera lenses covered with multiple layers of electrical tape as required by CREDO. A plan was in place to utilize as many devices as possible, and donations of old smartphones had already been received. The *Dangerous Goods Regulations* (DGR) by the *International Air Transport Association* (IATA) limits the number of lithium-ion powered devices; however, it also explicitly allows an airline to issue a permit to carry more if requested. Upon request, an agent of Lufthansa informed us that Lufthansa Airlines would not issue such a permit.

### 4.2. Experiment Implementation

#### 4.2.1. Application

The MRNG prototype has very few needs as it only requires Android API level 14. Alongside the visual representation of UHECR, current space weather information, as visible in Figure 1, was gathered with the MRNG prototype [65]. This information is presented in various forms, including images such as *OVATION Aurora Model Forecast North* [67,68,69], *Solar Wind Predicted at Earth Geospace Timeline Lates 24 h* [70], *Estimated Planetary K index (3 h data)* [71], and *Space Weather Overview incl. Solar X-ray Flux, Solar Proton Flux, Geomagnetic Activity* [72]. Moreover, there is text-based information available, such as *Aurora Hemispheric Power Tabular Values–Ovation Aurora Short Term Forecast* [73], *Daily Geomagnetic Data* [74], and *Solar Geophysical Event Reports* [75]. The information was collected in advance due to the availability and is depicted in Figure 1. A few more example datasets are shown within the Appendix C. Another reason for the data collection was to potentially explain unusual detection rate results that could have been caused by unpredictable solar particle events, such as the *Carrington Event in 1859*, events in August 1972, and events in October 1989 [48,76].

#### 4.2.2. Algorithm

Current research shows that muons can trigger a detection, hot-pixels, as well as background radiation [63]. On the one hand, radiation and general noise of the CMOS can trigger multiple pixels, whereas so-called hot pixels can trigger a single pixel. As per the research of Bar et al. [63], the authors indicated that focusing specifically on muons is currently the subject of research. We eliminated all static or noise elements, such as hot pixels, and most of the CMOS image sensor noises, with the following constraints:

Firstly, in order to eliminate CMOS background noise, we ignored all pixels where any of the RGB channels were beyond a certain threshold, epsilon. Secondly, to eliminate hot-pixel detections, images with more than two pixels beyond the set threshold were considered. We decided on 2 instead of 3 pixels as recommended by [63], as the post-processing of all collected detections eliminated all hot pixels that had more than 2 pixels fired with high certainty.

Moreover, the MRNG bit sequence was processed from four groups of input *time (P1)*, *position (P2)*, *color (P3)*, and *outlier (P4)*. It can be argued that P3 and P4 share many similarities; hence, only one can be used in the final random sequence. The abstract process of collecting the data is shown in Figure 2.

P1 Time.

The truncated last five digits of the Unix timestamp in milliseconds as a whole number converted into its binary representation. Hence, 100,000 possible numbers make a complete cycle every 100 min. By truncating leading zeros, the 1 to 0 proportion becomes more equally distributed; always starting with a 1 can be neglected, in our opinion, as the sizes of these sequences vary.

P2 Position.

Concatenation of the X-coordinate modulo 2 with the Y-coordinate modulo 2 results in 2 bits that contribute to the final random sequence.

P3 Color.

For each pixel where either the red, green, or blue color value (0–255) has a value beyond the pre-defined non-dynamic value epsilon, we calculate the modulo 2 of each color channel value, and add them together with the 2 other colors. The sum of all colors in the final step is, again, subjected to a modulo 2 calculation.

P4 Outlier.

Similar to P3, P4 is calculated in the event of a color channel reaching the predefined threshold. In addition, P4 is calculated only if the local distance between the previous color and the current color exceeds the overall average distance of the given color. The calculation is exactly the same as P3, where, firstly, the modulo 2 operation is executed on each color. Secondly, all values are added together, and the sum is again calculated with modulo 2. A code snipped showing the algorithm can be found in Appendix B.

### 4.3. Experiment Execution

The experiment period lasted for 25,990 min, from Wednesday, 16 March 2022 11:58:41.929 AM UTC+0 (1647431921929) to Sunday, 3 April 2022 1:08:35.353 PM UTC+0 (1648991315353); hence, it lasted for 18 days, 1 h, 9 min, and 54 s (18.05 days). The experiment was ultimately limited by the stay and physical presence within the Arctic 69° 40′ 53.117″ N 18° 58′ 36.027″ E at a 35 m elevation above sea level. The smartphones were checked regularly in order to verify the correct experiment execution.

## 5. Evaluation

We tested our MRNG generator within the Arctic Circle in Tromsø, Norway, at 35 m above sea level. The properties of the generated numbers were evaluated with the NIST SP.800-22 statistical test suite [35]. A few examples of the random sequences each hit generated are presented in Table 1. Column “P124L” shows that the number of bits captured the relevant hit for the most promising sequence “P124”, as discussed later in Section 6 and Section 7, respectively. The full dataset, including more columns, as well as secondary information on the current space weather during the detection of a hit, as shown in Figure 1, is available online [77].

**Table 1 entropy-25-00854-t001:** Excerpt showing 6 of the hits visible in Figure 3 alongside the extracted random sequences P1, P2, P3, and P4; the length of each sequence is indicated in the *P#L* column, and the overall length of the sequence from mode MRNG-P124 is shown in the P124L column.

Timestamp	P124L	P#L	P1–P4
1647655594901	52	17 2 34 33	**P1:** 10111001010110101 **P2:** 11 **P3:** 1101001010000101011001011010010001 **P4:** 110100101000010101100101101000001
1647670326687	46	15 2 29 29	**P1:** 110100000111111 **P2:** 11 **P3:** 11000011100101111010001111110 **P4:** 11000011100101111010001111110
1647889001366	49	11 2 44 36	**P1:** 10101010110 **P2:** 00 **P3:** 11100001101000111000101100001111100101100011 **P4:** 100011010011001011000011111001011001
1647992216262	37	14 2 27 21	**P1:** 11111110000110 **P2:** 00 **P3:** 111110011110101110110110111 **P4:** 110011110101110111011
1648193196369	34	17 2 17 15	**P1:** 10111100001110001 **P2:** 10 **P3:** 00100110111000010 **P4:** 001001011000010
1648302783786	28	17 2 11 9	**P1:** 10100011101001010 **P2:** 01 **P3:** 11111111110 **P4:** 111111110

### Detection

The experiment period lasted continuously for 25,990 min or about 18.05 days. During the experiment, the setup made 5567 uncleaned detections of hits throughout the 8 devices utilized. Under the assumption that hits were detected randomly across the whole image, these numbers indicate that a hit occurred, on average, about every 4.6667 min. By taking into consideration the size of the source image, the probability of a hit being at the same position can be expressed as 1/(height∗width), and seems rather unlikely. In the case of the 640×480 size image detection, the probability was 3.2552 × 10^−6^; this means that among 1,000,000 detections, it is likely that, on average, 3 detections share the same centered position on the image. In contradiction to this, Table 2 counts 3551 detections out of 3830 that share the same position. However, such detections are characterized as so-called *hot pixels* by the inventors of the forked CREDO application. Further, Homola et al. [10], Bar et al. [63] discuss the possibility that detections not exceeding 2 illuminated pixels are most likely not caused by cosmic radiation. Removing detections as stated above gives us a final set of 414 hits collected throughout 8 devices. This suggests that a hit caused by cosmic radiation can be expected every 62.7778 min over all 8 devices, on average.

Furthermore, as the experiment was being performed, a first look into the dataset showed that some detections were, to a great extent, beyond the predefined epsilon, as discussed in P3 Color Section within Section 4.2.2. An examination of the image revealed that the image in its entirety showed a tentative move toward red. This was confirmed by the RGB values throughout the image. For this reason, we implemented a trivial outlier detection form, which we named P4, as discussed in P4 Outlier Section within Section 4.2.2. During the evaluation of the data, we created a secondary blank white image. Pixels that were taken into consideration for P3 or P4 were painted black. With this method, we were able to cross-check our results visually. Figure 4, in the first row, depicts the hit (as collected by the application). The second row represents the used P3 pixels, and the last row represents the used pixels by P4.

In addition, we found that a previous approach of a simple threshold was not sufficient in all cases among all devices. Some detections were representations of a genuine muon but had all of the available 4096 pixels within a detection image beyond the defined threshold epsilon. The simple outlier categorization made it possible to extract only the genuine muons, as can be seen in the last row of Figure 4.

Finally, the extracted sequence was tested with the NIST SP.800-22 statistical test suite. Our obtained random sequence was not suitable to be tested by all 15 tests within the NIST SP.800-22 statistical test suite. However, as described in Section 3.3, we used all reasonable tests that were suitable for smaller sequences. This resulted in a set of six tests. The parameters used for the NIST SP.800-22 statistical test suite are described in Table 3.

## 6. Results

The settings and parameters used for the test execution can be found in Table 3. The results, as obtained from the NIST SP.800-22 statistical test suite, can be found in Table 4, Table 5, Table 6, Table 7, Table 8 and Table 9. The results were obtained from the file *‘finalAnalysisReport.txt’* within the folder structure of the NIST SP.800-22 statistical test suite. Tests that failed are marked with ‘*’ by the test suite. These results have an additional column called *‘PASS’*, which is not included within the results and only shows *‘YES’* if neither the *p*-VALUE nor the PROPORTION columns failed. In addition, failed rows are colored in red, whereas passing rows are colored in green, with the exception of the *‘STATISTICAL TEST’* column.

The extracted random sequences are tagged with their distinctive properties. ‘MRNG’, as a prefix, indicates the used RNG, followed by a hyphen ‘-’, followed by an *optional* ‘R’. to indicate raw unfiltered hits. A missing ‘R’ indicates that double hits and potential pixel errors are excluded from the sequence. This is followed by ‘P’ and integers, which clearly show which parts of the random sequence are included. For example, *’MRNG-RP1234’* includes P1 (Time), P2 (Position), P3 (Color), and P4 (Outlier) from all 5567 hits; whereas, *’MRNG-P124’* includes P1 (Time), P2 (Position), and P4 (Outlier) from the cleaned subset of 414 hits. Table 4, Table 5, Table 7, and Table 8 show the NIST SP.800-22 statistical test suite test results for the failed sequences MRNG-P1234, MRNG-P123, MRNG-RP1234, and MRNG-RP123. Whereas Table 6 and Table 9 show the results of passing sequences, namely MRNG-P124 and MRNG-RP124. Interestingly whenever a sequence includes the P3 features, the NIST SP.800-22 statistical test suite failed. This indicates that the P3 feature is a weak source of randomness; it shall be excluded from the final sequence. A table showing details on each sequence can be found within Appendix A.

Figure 5 and  Figure 6 demonstrate the asymptotic analytical outcomes of applying the Borel normality criterion [34] to both promising sequences, namely MRNG-P124 and MRNG-RP124. The relative frequencies of occurrences of bit patterns 0 and 1 should ideally be 50%. Furthermore, occurrences of each bit pattern of 2, such as 00, 01, 10, and 11, should be 25%, whereas bit patterns of 3 should match at 12.5%. Those values shall ideally be reached in an asymptotic manner [32,34]. An analysis of 414 MRNG-P124 sequences is shown in Figure 5. The analysis reveals that the occurrences of 0 and 1 are 48.97% and 51.03%, respectively. The sequences 00, 01, 10, and 11 are observed with occurrences of 24.46%, 24.28%, 26.36%, and 24.91%, respectively. Lastly, sequences 000, 001, 010, 011, 100, 101, 110, 111 had 12.18%, 12.22%, 12.07%, 12.38%, 13.23%, 13.00%, 12.67%, and 12.24%, respectively. Figure 6, on the other hand, fails to match that criterion in many occurrences.

The visualization highlights the average frequencies of various bit patterns found in these sequences. Notably, the error bars above each bar in the figure state the standard error derived from the corresponding standard deviations associated with the mean frequencies. This representation suggests that MRNG-P124 has an almost equal distribution of bit patterns within a given sequence. On the other hand, the chart representing the analysis of the unfiltered MRNG-RP124 sequences differs from the equilibrium distribution, suggesting that this sequence has a predictive pattern, hence holding weak randomness.

## 7. Discussion

By exploiting unwanted SEUs, we have successfully extracted a random sequence using UHECR as the entropy source. However, due to the properties of muons and the relatively low probability of UHECR detection, as discussed, we were not able to achieve a high bitstream rate, as needed for various cryptographic applications. By obeying the recommendations in the literature, as well as the documentation of the test suite itself, we were only able to feed 6 out of 15 tests provided by NIST SP.800-22 statistical test suite [35] with the extracted random sequence of our MRNG prototype. Nonetheless, our MRNG prototype passed all six of the selected NIST SP.800-22 statistical test suite tests with the sequences gathered through the *MRNG-RP124* and *MRNG-P124* methods. Moreover, a Borel normality criterion analysis on both methods strongly hints that the unfiltered *MRNG-RP124* sample sequences are not random. In the case of the *MRNG-P124*, a nearly equal distribution of bit patterns was evident, which looks promising to us.

One point to our advantage is that the MRNG prototype does not need the internet, a dedicated server, or a geolocation system, such as GPS. Despite our current MRNG prototype running only on Android smartphones, the technology used allows the MRNG prototype to be ported onto any system that provides a CMOS/CCD sensor and can execute code, thus rendering the MRNG independent. Space weather, as discussed in Section 3.4, can have a significant impact on the cosmic radiation measured on the surface of the Earth. For this reason, data from the NOAA Space Weather Prediction Center, as seen in Figure 1, were gathered and made available online along with the dataset [77]. The data on space weather during the detection of a hit has no impact on the random number sequence produced by the proposed MRNG but provides an opportunity for later analysis if anomalies are detected.

In Section 3.4, we discussed the inherently unpredictable nature of UHECR, specifically its sources, energies, and occurrences. We have shown that these characteristics can be harnessed to generate a source of randomness. The MRNG draws its randomness from the time of the interference, the position of the image sensor, and the effect on the image sensor from the interference with UHECR. Notwithstanding the limitations, especially on the short sequence, the study suggests that UHECR is a promising new entropy source for new RNGs. We see potential usage for our MRNG in areas where secure communication must be established by relying only on local and offline components, respectively, methods.

### 7.1. Splash-like Particle Representation

Figure 7 shows a splash-like representation of radiation, which, to the best of our knowledge, has not been previously mentioned. Homola et al. [10], Bibrzycki et al. [60], Bar et al. [63] only mention *trails*, *worms*, *artifacts*, and *spots*.

### 7.2. Muon Random Number Generator (MRNG)

UHECR can be detected and categorized using CMOS image sensors, commonly found in consumer smartphones or webcams [10,58,59]. The interference of muons with the CMOS image sensor, UHECR, seems to have a similar effect on CMOS/CCD image sensors to that of light. Physics research is very interested in the detection of muons, which are only part of UHECR; however, in our research, we did not differentiate between them. Despite this fact, in honor of their research and the genuinely fascinating muon, we propose the name *Muon Random Number Generator* (MRNG) for our proof of concept.

### 7.3. Reproducibility

The MRNG Android application is available on GitHub [65] and can be freely used and modified by other researchers to collect their own data. Our raw dataset, a processed database, and used helper scripts are freely available at Zenodo [77]. The data and applications can be used to confirm our results or use the data for further analysis.

### 7.4. Outlook

We believe that our theory to access UHECR as the entropy source by exploiting unwanted SEUs has shown promising results and it shall be researched further. In our opinion, the next step would be to find financial aid and scale the experiment up to the extent that all NIST SP.800-22 statistical test suite tests can be properly fed. This can be anywhere on Earth, as the location has, if any, a negligible impact. Furthermore, it is important to develop an open-source library for development that combines a further researched MRNG as well as other RNGs proposed by [9].

## Figures and Tables

**Figure 1 entropy-25-00854-f001:**
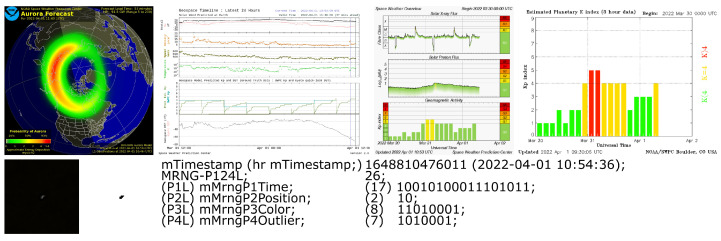
The first row represents fetched data from the NOAA Space Weather Prediction Center: *‘OVATION Aurora Model Forecast North’* [67,68,69], *‘Solar Wind Predicted at Earth Geospace Timeline Lates 24 h’* [70]—depicted in inverted colors, *‘Aurora Hemispheric Power Tabular Values—Ovation Aurora Short Term Forecast’* [73], *‘Estimated Planetary K index (3 h data)’* [71], whereas the second row shows the image of the hit, the P4 representation of the used pixel colored in black, and the textual data of the extracted random sequence.

**Figure 2 entropy-25-00854-f002:**
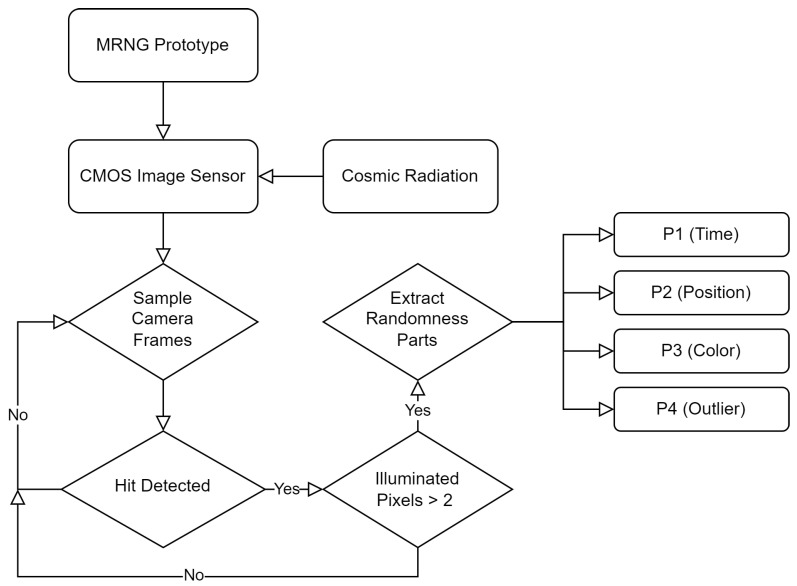
Depicts the abstract process of how the data are collected.

**Figure 3 entropy-25-00854-f003:**
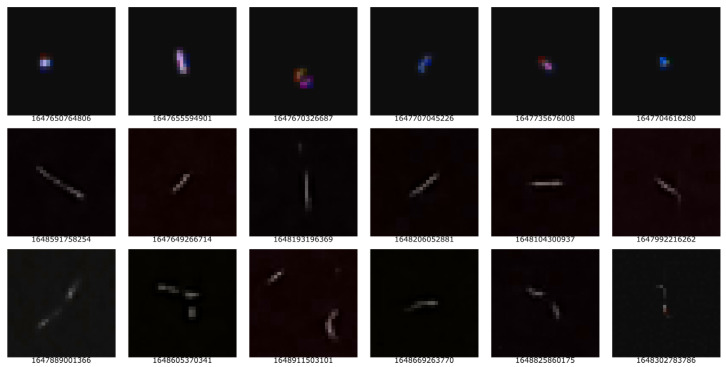
The hits of (presumably) UHECR and muons, respectively, cropped into a 32 × 32 frame, where the first row represents spots, the second represents tracks, and the third and last represent worms. The number beneath each 32 × 32 frame represents the UNIX timestamp when the hit is detected.

**Figure 4 entropy-25-00854-f004:**
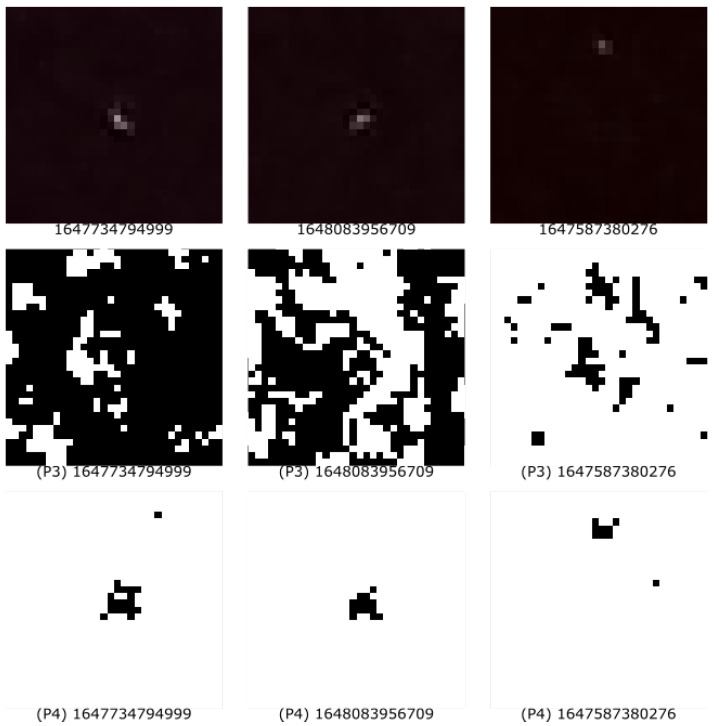
Comparison of the pixels taken into consideration for the random sequence from the visual representation of UHECR (first row) by feature P3 (second row) or feature P4 (last row).

**Figure 5 entropy-25-00854-f005:**
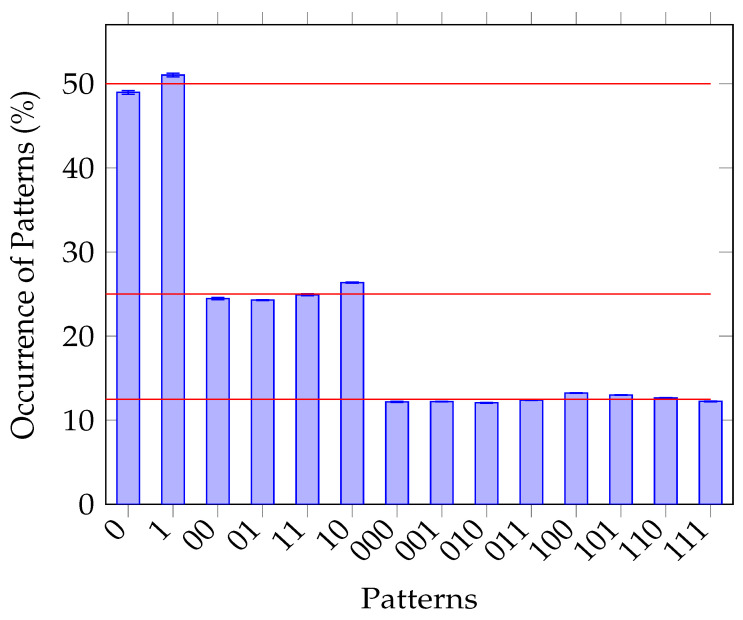
The results of the outcome from the Borel normality criterion [34] test. It analyzes the mean frequencies of bit patterns in 414 MRNG-P124 sequences (12 052 bits), with error bars representing the standard error. The red lines represent 12.5%, 25%, and 50%.

**Figure 6 entropy-25-00854-f006:**
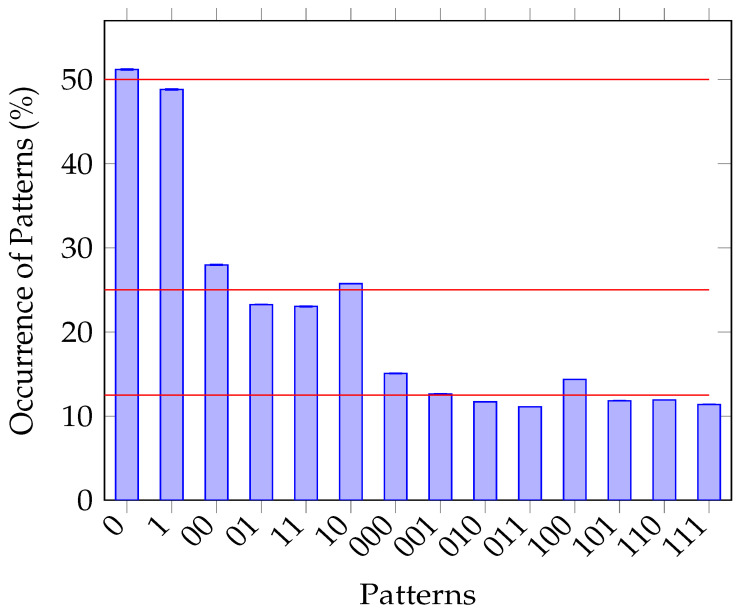
The results of the outcome from the Borel normality criterion [34] test. The illustration analyzes the mean frequencies of bit patterns in 5567 MRNG-RP124 sequences (126 363 bits), with error bars representing the standard error. The red lines represent 12.5%, 25%, and 50%.

**Figure 7 entropy-25-00854-f007:**
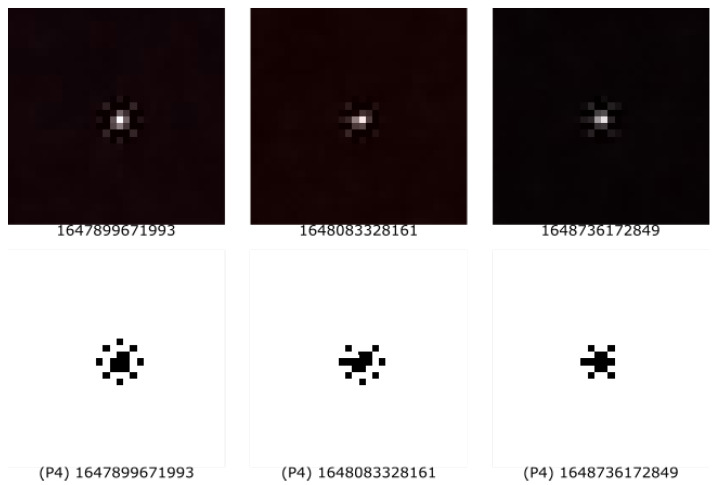
The figure displays the splash-like representation of (presumably) cosmic radiation found during the analysis of the dataset that has not, until now, been described or mentioned.

**Table 2 entropy-25-00854-t002:** Displays the number of double hits “cntDH” and programmatically detected potential pixel errors “cntPE”. The device identifiers (deviceID) were masked with asterisks as a full deviceID adds no value to this table but has a very small chance of being used in a malicious way.

deviceID	Width × Height	cntDH	cntPE
sm_a320fl-***14bd	640 × 480	2503	2157
sm_a320fl-***b4db	640 × 480	15	44
sm_a320fl-***1419	640 × 480	1033	16
sm_a505fn-***YVCM	1920 × 1080	1	752
mi_a1-***0804	1280 × 640	1	2

**Table 3 entropy-25-00854-t003:** Used parameters and settings to run the NIST statistical test suite.

Parameter	Value
Length of a bitstream	128
Number of bit streams	94
Applied statistical tests	1–5;11
Input file format	ASCII
Block frequency test—block length (M)	8
Approximate entropy test—block length (m)	2

**Table 4 entropy-25-00854-t004:** NIST  statistical test suite results for the random bit sequence **“MRNG-P1234”** from our MRNG prototype. Tests that failed are marked with ‘*’ by the test suite.

*p*-Value	Proportion	Statistical Test	Pass
0.000000 *	26/94 *	Frequency	NO
0.000000 *	49/94 *	BlockFrequency	NO
0.000000 *	28/94 *	CumulativeSums	NO
0.000000 *	28/94 *	CumulativeSums	NO
0.000000 *	50/94 *	Runs	NO
0.000000 *	38/94 *	LongestRun	NO
0.000000 *	31/94 *	ApproximateEntropy	NO

**Table 5 entropy-25-00854-t005:** NIST  statistical test suite results for the random bit sequence **“MRNG-P123”** from our MRNG prototype. Tests that failed are marked with ‘*’ by the test suite.

*p*-Value	Proportion	Statistical Test	Pass
0.000000 *	27/94 *	Frequency	NO
0.000000 *	59/94 *	BlockFrequency	NO
0.000000 *	30/94 *	CumulativeSums	NO
0.000000 *	30/94 *	CumulativeSums	NO
0.000000 *	53/94 *	Runs	NO
0.000000 *	44/94 *	LongestRun	NO
0.000000 *	30/94 *	ApproximateEntropy	NO

**Table 6 entropy-25-00854-t006:** NIST statistical test suite results for the random bit sequence **“MRNG-P124”** from our MRNG prototype.

*p*-Value	Proportion	Statistical Test	Pass
0.013153	92/94	Frequency	YES
0.000677	94/94	BlockFrequency	YES
0.189397	93/94	CumulativeSums	YES
0.804337	93/94	CumulativeSums	YES
0.100508	93/94	Runs	YES
0.332797	92/94	LongestRun	YES
0.879806	94/94	ApproximateEntropy	YES

**Table 7 entropy-25-00854-t007:** NIST  statistical test suite results for the random bit sequence **“MRNG-RP1234”** from our MRNG prototype. Tests that failed are marked with ‘*’ by the test suite.

*p*-Value	Proportion	Statistical Test	Pass
0.000000 *	26/94 *	Frequency	NO
0.000000 *	49/94 *	BlockFrequency	NO
0.000000 *	28/94 *	CumulativeSums	NO
0.000000 *	28/94 *	CumulativeSums	NO
0.000000 *	50/94 *	Runs	NO
0.000000 *	56/94 *	LongestRun	NO
0.000000 *	43/94 *	ApproximateEntropy	NO

**Table 8 entropy-25-00854-t008:** NIST  statistical test suite results for the random bit sequence **“MRNG-RP123”** from our MRNG prototype. Tests that failed are marked with ‘*’ by the test suite.

*p*-Value	Proportion	Statistical Test	Pass
0.000000 *	35/94 *	Frequency	NO
0.000000 *	52/94 *	BlockFrequency	NO
0.000000 *	33/94 *	CumulativeSums	NO
0.000000 *	34/94 *	CumulativeSums	NO
0.000000 *	55/94 *	Runs	NO
0.000000 *	49/94 *	LongestRun	NO
0.000000 *	37/94 *	ApproximateEntropy	NO

**Table 9 entropy-25-00854-t009:** NIST statistical test suite results for the random bit sequence **“MRNG-RP124”** from our MRNG prototype.

*p*-Value	Proportion	Statistical Test	Pass
0.000283	93/94	Frequency	YES
0.824517	93/94	BlockFrequency	YES
0.019334	93/94	CumulativeSums	YES
0.130453	92/94	CumulativeSums	YES
0.332797	93/94	Runs	YES
0.490050	93/94	LongestRun	YES
0.949602	94/94	ApproximateEntropy	YES

## Data Availability

The data [77] presented in this study are openly available at https://zenodo.org/record/7774330#.ZCGG2HbP2bh.

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
