# Peer review of "MRNG: Accessing Cosmic Radiation as an Entropy Source for a Non-Deterministic Random Number Generator"

_entropy, 2023, doi:10.3390/e25060854_

Round 1
Reviewer 1 Report
The manuscript uses the Ultra High Energy Cosmic Rays as the entropy Source to extract a random sequence successfully, by exploiting unwanted SEUs. The work provides evidence to some extent that the random bit sequence extracted from detections successfully pass established randomness tests. The study is somewhat interesting and appropriate in terms of non-deterministic random bit generator. The first three parts of the manuscript are written in detail, introducing entropy sources, attack models, existing research, testing methods, and etc. The experimental part mainly uses the NIST software package to test the randomness of the data, which is a relatively modest experimental record paper. The main problem is the lack of an independent theory or method.
Author Response
Thank you for the comments. Attached a PDF citing your comments and my response, marked in red.

Reviewer 2 Report
1. Some punctuation mistakes in lines 38, 115, and 131.
2: The authors did a nice historical review. But, they did not include the state-of-the-art (recent works) on systems that generate Random numbers, especially those encountered in optics and photonics that are nowadays dominant in literature. I strongly recommend the authors to include such an aspect.
3: Table 1 is not commented on in the text and Figure 1 is merely commented. What is the information displayed by the authors? Further comment is required.
4: A comment on how the authors collect the data is needed.
5. In the text (line 341), the authors argue that they were able to feed 6 out of the 15 NIST SP.800-22 statistical test suites. But, in tables 6 and 9, only 5 of the 15 NIST SP.800-22 statistical test suites passed.
6. In this manuscript, Random numbers are not generated since all of the NIST SP.800-22 statistical test suite must be passed to confirm that random numbers are generated. Moreover, the authors do not give the number of bitstreams used to validate the randomness. The author must do further tests with successful results.
Author Response

(The authors gave the same response as above.)

Reviewer 3 Report
-The paper presents a Muon Random Number Generator (MRNG) prototype that runs on Android smartphones and that doesn't require Internet, dedicated servers or geolocation systems. The MRNG prototype was used during an experiment and tested for its statistical strength.
-The approach uses Ultra High Energy Cosmic Rays (UHECR) embedded in a Random Number Generator (RNG) as a source of entropy.
-In many instances, reference to sources is not done correctly.
-The technical aspects of the research presented seem sound.
-It would have been useful for the reader to discuss how your results compare to baseline results or to results obtained with similar approaches.
-Considering the random nature of cosmic rays and the conditions in which you performed your experiment, which would be the best way for other researchers to replicate your experiment and/or compare results?
-What was the rationale of using 8 smartphones instead of 10 or more?
-It would have been good to include more details and a diagram to describe the methodology of the approach.
-You should have mentioned what is future work planned, if any, to continue developing your prototype.
-It would have been useful to include how would your approach could be applied in everyday life and how would it benefit people and/or organisations.
-I would recommend to have a detailed proofread of all the paper as there are some typos.
Author Response

(The authors gave the same response as above.)

Round 2
Reviewer 2 Report
No more comment
Reviewer 3 Report
I think the authors have addressed my comments and the comments of the other reviewers appropriately and in the best way they could. I think the manuscript has been sufficiently improved to warrant publication.